# Lipophosphonoxins—A Novel Group of Broad Spectrum Antibacterial Compounds

**DOI:** 10.3390/pharmaceutics15102395

**Published:** 2023-09-28

**Authors:** Viktor Šebek, Pavel Anzenbacher, Dominik Rejman, Alena Špičáková, Milan Kolář

**Affiliations:** 1Department of Pharmacology, Faculty of Medicine and Dentistry, Palacký University, Hněvotínská 3, 775 15 Olomouc, Czech Republic; sebek.viktor@gmail.com (V.Š.); pavel.anzenbacher@upol.cz (P.A.); 2Institute of Organic Chemistry and Biochemistry, Czech Academy of Sciences, Fleming Square 542/2, 160 00 Prague, Czech Republic; dominik.rejman@uochb.cas.cz; 3Department of Microbiology, Faculty of Medicine and Dentistry, Palacký University, Hněvotínská 3, 775 15 Olomouc, Czech Republic

**Keywords:** Lipophosphonoxins, LPPOs, antimicrobial resistance, antibiotics

## Abstract

Lipophosphonoxins (LPPOs) represent a new group of membrane-targeting antibiotics. Three generations of LPPOs have been described: First-generation LPPOs, second-generation LPPOs, and LEGO-LPPOs. All three generations have a similar mode of bactericidal action of targeting and disrupting the bacterial cytoplasmic membrane of prokaryotic cells, with limited effect on eukaryotic cells. First-generation LPPOs showed excellent bactericidal activity against Gram-positive species, including multiresistant strains. Second-generation LPPOs broaden the antibiotic effect also against Gram-negative bacteria. However, both first- and second-generation LPPOs lose their antibacterial activity in the presence of serum albumin. LEGO-LPPOs were found to be active against both Gram-positive and Gram-negative bacteria, have better selectivity as compared to first- and second-generation resistance to LEGO-LPPOs was also not observed, and are active even in the presence of serum albumin. Second-generation LPPOs have been studied as antimicrobial additives in bone cement and as nanofiber dressing components in the treatment of wound infections in mice. Second-generation LPPOs and LEGO-LPPOs were also tested to treat ex vivo simulated endodontic infections in dental root canals. The results of all these studies were encouraging and suggested further investigation of LPPOs in these indications. This paper aims to review and compile published data on LPPOs.

## 1. Introduction

Ever since penicillin started to be widely used for treating soldiers during World War II, antibiotics started to be considered wonder drugs with the potential to save millions of lives, along with the introduction of better hygiene [1,2]. The span between the years 1945 and 1965 is called the golden era of antibiotics, and as many as half of the antibiotics commonly used today were discovered in this period [3]. Increased usage, and especially overuse and misuse of antibiotics in the health care settings and agricultural industry, has led to the development of bacterial pathogens resistant to some antibiotics. It is well known that Sir Alexander Fleming, the man who discovered penicillin in 1928, had already warned the public in his Nobel lecture in 1945 about the future high demand for antibiotics and their overuse that could lead to the emergence of resistance. “*It is not difficult to make microbes resistant to penicillin in the laboratory by exposing them to concentrations not sufficient to kill them, and the same thing has occasionally happened in the body. The time may come when penicillin can be bought by anyone in the shops. Then there is the danger that the ignorant man may easily underdose himself and by exposing his microbes to non-lethal quantities of the drug make them resistant*” [4]. Today, the accelerated emergence of antimicrobial resistance (AMR) is posing a huge public healthcare as well as an economic issue not only in underdeveloped countries but globally [5]. According to the European Centre for Disease Prevention and Control, every year, more than 35,000 people die due to antimicrobial-resistant infections in the EU/EEA countries. The death toll is comparable to that of influenza, tuberculosis, and HIV/AIDS combined [6]. Globally, antimicrobial-resistant bacteria kill nearly 1.3 million people a year [7], and in the USA alone, over 2 million ailments per year are caused by multiresistant bacteria with an estimated treatment cost range of 20–35 billion USD [8]. AMR is becoming one of the main public health problems of this century, and in the current scenario, the death toll could rise to 10,000,000 by 2050 [9].

Many currently available and used antibiotics target five biosynthetic processes that occur in actively growing bacteria: the biosynthesis of proteins, RNA, DNA, peptidoglycan, and folic acid. However, most of these modes of action are not effective against persistent bacteria, and bacterial strains resistant to antibiotics are emerging [10]. Because of resistance development, the search for new group/s of antibiotics that will be active against resistant bacteria has attracted an enormous effort in the pharmaceutical industry and by research groups in academia. About 20 Big Pharma companies joined forces with the public sector (e.g., the World Health Organization and the European Investment Bank) and founded AMR Action Fund, which plans to invest 1 billion USD into AMR research with the aim to discover two to four new antibiotics by 2030 [7].

Since there is a fundamental difference in the composition of the cytoplasmic membrane between the bacterial/prokaryotic and the mammalian/eukaryotic cells, the membrane is becoming an attractive target for the development of new antibacterial agents. Agents targeting the bacterial cytoplasmic membrane have the potential to kill multiresistant and dormant (non-multiplying) bacteria [11]. There are many membrane-active compounds under development, and some are already in clinical practice (e.g., colistin and daptomycin).

Antimicrobial peptides (AMPs) are part of the innate immunity of essentially all species investigated, ranging from insects to humans. AMPs have a wide range of inhibitory effects against bacteria, fungi, parasites, and viruses and are extensively investigated [12]. AMPs are large molecules, ranging from 12 to 50 residues, and have an affinity for the negatively charged prokaryotic membranes. They have broad-spectrum antimicrobial activity, interact with the membranes, and increase their permeability to ions and solutes that may lead to cell death [13]. Rapid disruption of microbial cell membranes makes bacteria find it difficult to develop resistance. However, peptide antibiotics showed relatively little clinical success due to their in vivo toxicity, short half-lives, low in vivo stability, limited tissue biodistribution, and large production costs. Their use is generally limited to intravenous administration or local applications [12,14].

To overcome the limitations of AMPs, research activities focused on peptidomimetics that still have the biophysical characteristics of AMPs. However, they are relatively simple to synthesize (smaller molecules than AMPs), and they have longer half-lives in vivo. Peptidomimetics are under clinical development as they demonstrated activity against a broad spectrum of Gram-positive bacteria (e.g., LTX-10) [15] as well as against Gram-negative bacteria (e.g., ceragenin CSA-13) [16,17]. Positively charged peptidomimetics are electrostatically attracted to the negatively charged bacterial membranes. This interaction works less effectively in eukaryotic cells because they contain neutral lipids (i.e., neutral charge) in the outer leaflet of their cytoplasmic membrane, hence the higher affinity of peptidomimetics for the prokaryotic cells.

## 2. Lipophosphonoxins

In the golden age of antibiotic discovery, the cytoplasmatic membrane was not considered a therapeutic target due to concern that the mammalian cells could be damaged by the antibiotics targeting the bacteria [18]. This view, however, is changing. Lipophosphonoxins (LPPOs) belong to a small molecule antibacterial membrane targeting peptidomimetics. Hence they target the cytoplasmic membrane. This new and very promising class of antibiotics was introduced for the first time in 2011 by Rejman et al. in the Journal of Medicinal Chemistry [19]. At that time, the LPPO group that was investigated was called the first-generation today, as today there is already a second and a third generation of LPPOs, which seems to be even more promising for clinical use than the first one. The third generation is also termed LEGO-LPPOs [14].

### 2.1. First-Generation Lipophosphonoxins

In general, LPPOs are a heterogeneous group with a similar structural pattern. All LPPOs are molecules built from four parts: a nucleoside module, an iminosugar module, a hydrophobic module (lipophilic alkyl chain), and a phosphonate linker module that holds the first three modules together [19] (see Figure 1). LPPOs‘ modular structure allows a few steps of reaction synthesis of different compounds from this group with different properties, such as increasing their selectivity and extending their antibacterial activity. All LPPOs are amphiphilic. It means that they contain both a lipophilic and a hydrophilic part. First-generation LPPOs have hydrophilic moieties with a small positive charge [20].

Antimicrobial activity of the first-generation LPPOs was investigated in Gram-positive and Gram-negative bacteria, including multiresistant bacterial strains (e.g., vancomycin-resistant *Enterococcus faecium* and methicillin-resistant *Staphylococcus aureus*). In vitro studies demonstrated that the first generation of LPPOs has selective cytotoxicity against Gram-positive bacteria, and the most active compounds were also active against multiresistant strains. The lack of efficacy against Gram-negative bacteria is likely due to LPPOs’ inability to cross both bacterial membranes, the outer membrane and the cytoplasmatic membrane. It was also demonstrated that configuration at the phosphorus atom in the LPPO molecules does not play an important role in the antibacterial activity [19]. The antibacterial activity differed among different investigated first-generation compounds, but the minimum inhibitory concentration (MIC) values of the best inhibitors were in the range of 1–12 mg/L. The most promising ones selected for further investigations were DR-5047 and DR-5026 [20].

The antibacterial effect of DR-5047 and DR-5026 was tested on the biosynthesis of DNA, RNA, proteins, cell wall peptidoglycan, and lipids, and concluded that LPPOs did not significantly affect the biosynthesis of any of these macromolecules. The mode of antibacterial action goes through the damage of the bacterial cytoplasmic membrane. LPPOs accumulate on both sides of the phospholipid bacterial membrane bilayer and disrupt it by forming pores in it (see Figure 2). The presence of the pores leads to an efflux of the bacterial cytosol out of the cell and, ultimately, after the longer incubation with LPPOs, to complete disintegration of the cell [20].

The pore formation was specific for prokaryotic membranes; the effect of LPPOs on eukaryotic membranes (tested on human erythrocytes and macrophages) was less pronounced. Whereas prokaryotic membranes have negatively charged lipids and cholesterol is universally absent there, the eukaryotic membranes are composed of phospholipids with neutral charge and cholesterol, which increases the membrane thickness and decreases its fluidity, is present there [21]. This structural difference between the membranes explains why LPPOs don’t cross the eukaryotic plasmatic membrane at their bactericidal concentrations and can’t create pores in it.

Bactericidal concentrations of LPPOs were detected in the bacterial cytoplasmic membrane but not in the membranes of the eukaryotic/human cells. The cytotoxic concentrations against human cell lines were significantly above the range of MIC (1–12 mg/L), and IC_50_ (Half maximal inhibitory concentration) was in the range of 60–100 mg/L [20].

Neither DR-5047 nor DR-5026 compound showed to be genotoxic at MIC concentrations (determined by the Ames test on two bacterial strains of *Salmonella* Typhimurium) [20]. The absence of genotoxicity of LPPOs is associated with their inability to enter the cytosol.

LPPOs were investigated in membrane permeabilization in mouse macrophages and showed not to affect the cytoplasmic membrane permeability of the eukaryotic cells. Transepithelial transport of LPPOs was tested in Caco-2 monolayers and showed that neither compound passed through the Caco-2 monolayer or was absorbed by the cells. It suggests the inability of LPPOs to be transported through the epithelium and absorbed in the intestine of mammals. On top of that, LPPOs demonstrated excellent stability at low pH values, regardless of temperature, with no detectable decomposition. This suggests that if LPPOs are administered orally, they can pass undamaged through the acidic environment of the stomach [20].

Apart from the activity against the Gram-positive bacteria, the maximum tolerated dose (MTD) and the potential for the development of bacterial resistance against LPPOs were investigated. The DR-5026 compound was tested for MTD in mice, and even the highest dose tested (2000 mg/kg) did not show any signs of toxicity when administered perorally. However, when administered intraperitoneally, the safe limit was significantly lower (50 mg/kg), which makes first-generation LPPOs unsuitable for systemic application [20]. On top of that, in vitro tests showed that the antibacterial activity of first-generation LPPOs was lost in the presence of bovine serum albumin; hence, toxicity is not the only limiting factor hindering systemic treatment. The potential for the development of resistance against LPPOs was tested in *Enterococcus faecalis* and *Streptococcus agalactiae* strains with subinhibitory concentrations of LPPOs. MIC was determined after each of the 14 passages and stayed unchanged. Furthermore, the development of resistance was tested in *Bacillus subtilis* with subinhibitory concentrations of either DR-5026 or rifampicin. After a couple of cycles of 24 h incubation and an increase in concentration of the respective inhibitory compound, there were developed strains growing at concentrations exceeding MIC of rifampicin; however, no strains growing at concentrations exceeding MIC of DR-5026 [20]. It was demonstrated that LPPOs have a relatively low propensity for resistance development, probably due to their membrane-targeting mode of action, which makes this group of compounds attractive for further development.

### 2.2. Second-Generation Lipophosphonoxins

While first-generation LPPOs demonstrated selectivity against prokaryotic cells, showing no or limited genotoxicity against human cell lines at minimal bactericidal concentration, having a mode of action that kills the prokaryotic cells by pore formation in their cytoplasmic membrane, and having a low propensity for resistance development, first-generation LPPOs are active only against Gram-positive pathogens and are unsuitable for systemic treatment. To overcome these limiting factors of first-generation, second-generation was synthesized and tested.

As mentioned earlier, the molecule of LPPO comprises four different modules that can be modified, and by these modifications, different LPPO agents can be created. The lack of susceptibility of Gram-negative bacteria to first-generation LPPOs is probably caused by the limited LPPO–membrane interaction due to a weak positive charge of the first-generation. As the positive charge in the first-generation structure is located on the imino-sugar module, the structural modification for the second generation was created here. Such a re-design of the imino-sugar module increased the number of positive charges and thus increased the affinity to the cytoplasmic membrane of the Gram-negative bacteria (see Figure 3).

The second-generation LPPOs demonstrated in vitro an increased efficacy against Gram-positive pathogens vs. first-generation (MIC < 1–6 mg/L) as well as antibacterial activity against Gram-negative pathogens, including clinically relevant strains of *Escherichia coli*, *Pseudomonas aeruginosa*, and *Salmonella* Enteritidis [22]. Both generations of LPPOs share the same mode of action—the creation of pores in the cytoplasmic membrane, resulting in the efflux of the bacterial cytosol and cell disintegration. As in the first-generation, second-generation showed no effect on the eukaryotic cells at their bactericidal concentrations, no in vivo inhibition of biosynthesis of the cell macromolecules (DNA, RNA, protein, peptidoglycan, and membrane lipids), excellent thermostability and stability in low pH, and inability to pass through the Caco-2 monolayer.

Of all synthesized second-generation LPPOs, DR-6155 and DR-6180 were selected for further investigation. Because of the inability of LPPOs to cross the eukaryotic cytoplasmic membrane and recommended local gastrointestinal and topical/skin application, LPPOs’ in vivo toxicity was investigated accordingly. The Maximum Tolerated Dose (MTD) was investigated in mice using an orally administered dose of 2000 mg/kg of body weight. No death, body weight loss, or gross pathology changes were observed with either compound during a period of two weeks [22]. Skin irritation test was performed in rabbits and lasted for at least one week. All tested animals were without any morphological, physiological, or behavioral abnormalities [22]. All these tests suggest a favorable safety profile of second-generation LPPOs in mammals.

One of the biggest assets of the first generation was a low propensity for resistance development, and thus resistance development was also investigated in the second generation. Twenty one-day passages with subinhibitory concentrations were carried out in vitro with *Pseudomonas aeruginosa* and ciprofloxacin as a control. The MIC remained relatively unchanged for LPPO and significantly increased for ciprofloxacin (from 0.25 to 4 mg/L). No *Pseudomonas aeruginosa* cells resistant to second-generation LPPOs were cultivated, but cells resistant to ciprofloxacin emerged [22]. The low propensity for pathogen resistance was shown also in Gram-negative bacteria.

Second-generation LPPOs keep all the beneficial properties of first-generation and, in addition, are effective against a broad spectrum of bacteria, including bacterial strains that are difficult to kill—Gram-positive as well as Gram-negative ones. Therefore, the logical step was to investigate further only the second generation. One of the few communicated disadvantages so far has been the unsuitability of LPPOs for systemic treatment, not only because of their inability to be absorbed from the intestine (that can be overcome by intravenous administration) but because of the interaction with the serum albumins that abolish the LPPOs’ antibacterial activity [14]. Hence, the second generation was investigated in the non-systemic applications—skin infections, as antimicrobial additives in bone cement, and in tooth root canal infection treatment.

#### 2.2.1. Second-Generation LPPOs as Antimicrobial Additives in Bone Cement

Infections are serious complications associated with 1–2% of primary joint replacements and may result in the development of bacterial biofilms on the implant surface that can result in implant removal [23]. About two-thirds of orthopedic implant infections are caused by staphylococci, and the second most common pathogens are streptococci [24]. To prevent the formation of bacterial biofilms, gentamicin is the most commonly used antibiotic mixed with poly(methyl methacrylate) (PMMA) bone cement. However, some bacterial strains are already resistant to gentamicin, and therefore, there is a need to find more potent antibiotics that can be mixed with bone cement without compromising the cement’s mechanical properties that would be released from the cement in sufficient amounts to prevent biofilm formation. Second-generation LPPOs, DR-6155 and DR-6180, were evaluated as additives to surgical bone cement in a series of in vitro studies.

During the polymerization process, exothermic reaction increases the temperature of the PMMA bone cement up to 90 °C [25]; hence LPPOs’ thermostability at high temperatures was evaluated. Both tested LPPOs were dissolved in 80 °C water for up to 8 h, although the polymerization takes only about 10 min. Then, LPPOs were tested with liquid chromatography–mass spectrometry, which showed no degradation and proved excellent thermostability of LPPOs. This is essential for withstanding high temperatures associated with surgical bone cement polymerization [26].

Another experiment tested whether adding LPPO into bone cement changes the characteristics of the material, such as strength and elongation, at break. The experiment demonstrated that composite cements containing up to 0.2 g of LPPO/10 g of cement did not negatively affect the polymerization process, and cement containing LPPOs had similar qualities as compared to one without LPPOs.

Investigators also tested whether LPPOs are released from the polymerized cement in an active form and in a sufficient amount. LPPO amounts ranging from 0.05 g to 0.2 g per 10 g of the PMMA bone cement were investigated, and the 0.18 g/10 g combination was selected as the one with the optimal LPPO release and still not compromising the mechanical properties of the bone cement [26]. LPPO is released in two phases. First comes an initial spike that lasts for a few hours and is followed by a gradual lower release in subsequent days. This biphasic release is optimal for local infection treatment when elevated initial concentration kills present bacteria, and subsequently released lower dose still suffices to prevent new bacterial infection from occurring [26].

The final test was designed to answer the question of whether LPPOs released from the bone cement can prevent bacterial biofilm formation. Bacterial strains of *Enterococcus faecalis*, *Staphylococcus aureus*, *ica* operon-positive *Staphylococcus epidermidis* (*ica* operon, a gene cluster encoding the production of intercellular adhesin, a polysaccharide that mediates the intercellular adherence of bacteria and biofilm accumulation), *Escherichia coli*, and *Pseudomonas aeruginosa* were investigated. Gentamicin (0.9 g per 40 g of PMMA bone cement) was used as a control. DR-6155 demonstrated complete inhibition of the biofilm formation for all tested bacterial strains, and DR-6180 showed complete biofilm inhibition for all strains except for *Pseudomonas aeruginosa*. Gentamicin prevented biofilm formation for all strains, with the exception of gentamicin-resistant *Staphylococcus epidermidis* [26].

#### 2.2.2. Nanofiber Dressing Loaded with Second-Generation LPPO in Treatment of Wound Infection Induced by *Staphylococcus aureus*

Systemic antibiotic therapy is the standard of care for the treatment of wound infections, especially burns, but it may be associated with systemic toxicity and/or with the emergence of resistant bacterial pathogens. Topical antibiotic therapy rarely causes systemic toxicity; however, it may have cytotoxic effects on keratinocytes and fibroblasts and thus delay wound healing [27]. Conventional wound dressings such as gauze do not prevent infection and drying of the wound beds. In addition, it needs to be replaced regularly, and this process often damages the healing tissue. From this perspective, the ideal wound dressing would be biodegradable, would support gas exchange, and would have antibacterial activity at the same time.

One of the most promising wound dressing systems is a polycaprolactone (PCL)-based nanofibrous scaffold (NANO) produced by electrospinning. NANO mimics, with its network of cross-linked smooth fibers, the structure of the native extracellular matrix (diameter of fibers around 1 μm and pores of several micrometers). It delivers oxygen, extracts wound exudate, and is biocompatible and biodegradable. It degrades into naturally occurring metabolite 6-hydroxyhexanoic acid and so reduces the rate of dressing replacements [28]. PCL scaffold is approved by the FDA for biomedical applications.

To enhance the NANO wound dressing properties and for the purpose of this experiment, NANO was loaded with second-generation LPPO DR-6180, and investigators tested whether LPPO is released from NANO and if so, whether it has antimicrobial activity against *Staphylococcus aureus*, whether it affects proliferation, differentiation, and migration of fibroblasts and keratinocytes, and whether LPPO is absorbed and has any systemic effect. NANO with loaded LPPO of concentrations 0, 2, 5, and 10% was used. NANO loaded with LPPO did not significantly affect the fiber size.

For the proper antibiotic function of the NANO–LPPO, it is crucial that LPPO is released in a sufficient amount. LPPO release was tested in a phosphate-buffered saline (PBS) environment and showed a dose-dependent pattern. LPPO from the NANO–LPPO10% was released rapidly in the first 24 h, followed by continuous slower release over the next six days. However, there was no detectable release from NANO–LPPO2% observed during the seven-day incubation period in the PBS. It suggests that the release of LPPO by simple diffusion occurs only at higher LPPO concentrations. Because lipases are ubiquitous enzymes produced by bacteria, in the next step, LPPO release from the NANO was tested in the presence of a lipase. In this environment, the release was observed even from NANO–LPPO2% (75% of LPPO released during the first 24 h, followed by slower release afterward). All the LPPO from NANO–LPPO10% was released, and the NANO was degraded during the first 24 h (50% during the first 8 h). The degradation of the nanofibers by lipase started as surface modifications and continued from day two in significant restructuring of the entire fiber, and the degradation was more pronounced with the higher LPPO concentration load. Interestingly, and this phenomenon was not described before, the degradation of the nanomaterial is accelerated not only by the presence of LPPO but also by the microbial lytic enzymes. It means that more bacteria produce more enzymes (lipases), and this causes bigger and faster NANO degradation with more LPPO released. Overall, LPPO is released from the NANO by two main mechanisms: simple diffusion and the enzymatically catalyzed degradation of NANO [29]. LPPO biphasic release, described as an initial spike within a few hours followed by a gradual lower release in subsequent days, is optimal for local infection treatment when elevated initial concentration kills present bacteria and subsequently released lower dose still suffices to prevent bacterial inhibition.

Wettability is defined as the ability of a liquid to spread over a surface, and the wettability of the dressing is an important characteristic that provides information about the ability of the dressing to absorb the fluids. The Washburn adsorption test showed that the presence of LPPO in the nanomaterial increased its wettability as compared to NANO without an LPPO load. LPPO-loaded NANO increases the ability to absorb the fluids and thus control the moisture in the wound [29].

Fibroblasts and keratinocytes are two of the major cell types responsible for the wound regeneration process, and their impairment by antibiotics may have a negative effect on wound healing. Therefore, it was investigated in vitro whether LPPO at bactericidal concentration has any effect on human dermal fibroblasts (HDFs) and immortalized human keratinocyte cells (HaCaT), and whether LPPO interferes with TGF-1 signaling, which is important for proper wound closure. It was demonstrated that LPPO in the concentration range from 0.1 to 25 mg/L does not impair the function of fibroblasts and keratinocytes. However, higher LPPO concentrations of 50 and 100 mg/L were found to be toxic to them. The keratinocyte migration, which is crucial for proper wound re-epithelization, is accelerated in the presence of pro-fibrotic cytokine (TGF-1β). In vitro studies showed that LPPO did not interfere with TGF-β1 signaling and thus did not alter the ability of keratinocytes to migrate [29].

The positive impact of nanomaterials on wound healing has been published before. It was proved that nanofibrous scaffolds hindered bacterial growth and thus promoted wound repair [30]. The effectiveness of NANO-LPPO dressing was investigated in non-infected as well as *Staphylococcus aureus*-induced wound infection in mice as *Staphylococcus aureus* impairs the wound healing process. In non-infected mice wounds, the NANO-LPPO dressing did not impair wound healing at any of the tested LPPO concentrations and even slightly improved granulation tissue formation. In *Staphylococcus aureus*-infected mice wounds, increased re-epithelization, granulation tissue formation, and newly formed collagen in the granulation tissue were described with NANO-LPPO5% and NANO-LPPO10% dressing, and no differences were observed as compared to non-infected control. These histologically infected wound healing properties of 5% and 10% NANO-LPPO concentrations were almost not observed with NANO-LPPO2%, where the histological wound picture looked similar as compared to infected wounds with NANO dressing without LPPO (see Figure 4) [29].

The higher positive impact of NANO-LPPO5% and 10% on infected wound healing was translated into an inspection of bacteria from wound swabs grown on agar media and *Staphylococcus aureus* qPCR-based quantification in wounds. On agar media, *Staphylococcus aureus* was detected in all samples without LPPO and with NANO-LPPO2% dressing; however, only in one sample with LPPO5%. In qPCR-based quantitation that also detects DNA released from dead bacteria, the bacterial load from wounds and surrounding skin treated with NANO–LPPO5% and 10% was significantly lower versus that from wounds treated with NANO2% and NANO only (in the range of 10^4^ vs. 10^6^, respectively) [29].

The report of Do Pham et al. [29] was the first one that described, at least partially, LPPO absorption when used locally and subsequent systemic distribution in the blood and liver of a mammal (mice in this case). In NANO–LPPO2%, the concentration of LPPO detected in mice plasma was 1.62 ng/mL in the non-infected mice and 3.39 ng/mL in the infected mice. In NANO–LPPO5%, the plasma concentration was 3.96 ng/mL and 1.58 ng/mL, respectively. In NANO–LPPO10%, the plasma concentration was 9.56 ng/mL and 8.49 ng/mL, respectively. In NANO–LPPO2%, the concentration of LPPO detected in the liver was 113.8 ng/mL in the non-infected mice and 72.23 ng/mL in the infected mice. In NANO–LPPO5%, the liver concentration was 222.5 ng/mL and 127.9 ng/mL, respectively. In NANO-LPPO10%, the liver concentration was 659.2 ng/mL and 161.7 ng/mL, respectively. There were three mice in each group. Overall, the concentration of LPPO detected in blood and liver correlated with its amounts in the NANO–LPPO dressing, and the systemic exposure was negligible from the cytotoxicity point of view [29].

### 2.3. LEGO–Lipophosphonoxins

Second-generation LPPOs showed a broader antibiotic spectrum than the first-generation ones while keeping all the other positive attributes of the first generation. However, neither generation is suitable for systemic treatment because serum albumin abolishes the LPPOs’ antibacterial activity. Therefore, the second generation was investigated only in non-systemic applications, skin infections, and as bone cement antimicrobial additives. To overcome the limitation of second-generation LPPOs, structure–activity relationship (SAR) studies were performed aiming to synthesize the next-generation LPPO that would keep excellent antimicrobial effects even in the presence of serum albumin and thus make it suitable for systemic treatment. Modifying various modules of the second-generation LPPOs (e.g., the nucleoside module) did not lead to significant improvement. Although some of the synthesized compounds retained their antibacterial activity, this ability was also lost in the presence of bovine serum albumin [14]. Thus, a new structural scaffold termed LEGO-LPPO was introduced, and if third-generation LPPO collocation is used, it refers to LEGO-LPPOs today.

The structure of LEGO–LPPOs is not complicated and can be easily synthesized in a few steps. The central phosphonate linker module (one of four modules) of LPPO is linked with two attached connector modules on either side, and each connector module is linked with one polar and one hydrophobic module (see Figure 5). Hence, these compounds were termed linker-evolved-group-optimized LPPOs, in short, LEGO–LPPOs [14].

Antibacterial activities of LEGO–LPPOs were evaluated in Gram-positive and Gram-negative bacterial pathogens, including the resistant strains, and hemolytic activity (HC_50_) was also tested. The most effective LEGO–LPPOs showed excellent MIC ranging from <1 to 8 mg/L, which was even lower than MIC observed with the second generation. High selectivity of LEGO–LPPOs is demonstrated by very high hemolytic concentration against erythrocytes that were at least 20–100 times above their respective MIC values [14] (versus HC_50_ values in the range of 16–30 mg/L with second-generation).

Out of more than 70 LEGO–LPPOs synthesized, the most effective ones were tested on 24 strains of wild-type and methicillin-resistant *Staphylococcus aureus* strains and compared with clinically used antibiotics, namely with penicillin, oxacillin, ampicillin/sulbactam, chloramphenicol, erythromycin, clindamycin, ciprofloxacin, and gentamicin. While each strain of tested pathogens was resistant to at least one antibiotic, LEGO–LPPOs showed efficacy against all strains with no sign of predisposed resistance [14].

In vitro experiments with *Escherichia coli* and *Staphylococcus aureus*, it was demonstrated that LEGO–LPPOs kill all tested bacterial strains within several hours, and the effect was concentration-dependent. With *Escherichia coli*, *Pseudomonas aeruginosa*, and *Staphylococcus aureus*, it was demonstrated that LEGO–LPPOs are also effective against persistent strains [14]. The efficacy is superior to ampicillin/sulbactam in all cases, equal to colistin in *Escherichia coli*, and less effective than colistin in *Pseudomonas aeruginosa* and daptomycin in *Staphylococcus aureus* [14].

LEGO–LPPOs are effective against Gram-positive as well as Gram-negative pathogens; however, there are significant differences among the compounds. The hydrophobicity index of the compounds seems to correlate with their activity and selectivity. While the least hydrophobic compounds showed no or very low antibacterial activity, the compounds exhibiting high lipophilicity showed low selectivity and high hemolytic activity. In between those two groups, there are compounds with broad-spectrum activity and high selectivity [14], and these are promising for further development into potential therapeutics. The most promising ones so far are ddp117_1 and DR-7072 (compounds numbered 25 and 38, respectively, in reference [14]).

One of the overarching characteristics across all generations of LPPOs was the difficulty in developing the resistant bacterial strain, and none were observed during the experiments, so compounds ddp117_1 and DR-7072 were also evaluated for developing *Pseudomonas aeruginosa* resistance. No resistant strains were cultivated with LEGO–LPPOs; however, resistant strains were selected with control antibiotics ciprofloxacin and ceftazidime [14].

There is also no significant difference in the mode of action of LEGO- and the other generations of LPPOs. In vitro tests with LEGO–LPPOs showed the ability to rapidly depolarize bacterial membranes in *Staphylococcus aureus* and *Escherichia coli* within a few seconds. However, the loss of the membrane potential did not kill the cells immediately (within hours). The mode of antibacterial action was the same as described in first- and second-generation. It means damage to the bacterial phospholipid bilayer and disrupting it by forming pores in it [14]. It suggests that LEGO–LPPO causes cell metabolic arrest first as the membrane potential is important for generating energy, active metabolism, and cell division [31], and LPPO slowly kills the cells by pore-forming activities only then.

The biggest and the most beneficial difference between LEGO–LPPOs and their first- and second-generation predecessors is in their antibacterial activity in the presence of bovine serum albumins. First- and second-generation activity was abolished by the albumin, and hence, those generations were not investigated in systemic treatment. In contrast, the antibacterial activity of LEGO–LPPOs is only slightly, if at all, affected by albumin [14], and thus LEGO–LPPOs have the potential to be developed for systemic treatment.

The compounds numbered ddp117_1 and DR-7072 were tested for cytotoxicity on mammalian cell cultures (HepG2 cells) and in skin and eye irritation tests to evaluate their safety as potential therapeutics. The cytotoxic concentration of both compounds was significantly above the MIC values. In vitro skin and eye irritation tests with concentrations of 20 and 200 mg/L showed no detrimental effects. The in vivo skin and eye irritation test was performed with concentrations of 100 and 200 mg/L on skin and with concentrations of 100 mg/L on rabbits’ eyes. Both tests showed no adverse effects on the skin and eyes of the mammals [14].

As a first step to test the systemic use of LEGO-LPPO, the compound numbered 25 was also assessed for maximum tolerated dose (MTD) in mice for oral and subcutaneous (s.c.) administration. MTD for oral administration was determined as >200 mg/kg of body weight, and the one for s.c. administration, it was >15 mg/kg of body weight [14]. Very importantly, no negative clinical signs were detected in living mice, and no significant pathological changes were found during the autopsy.

#### Second-Generation and LEGO–LPPOs Ex Vivo Effect on Root Canal Biofilm Produced by *Enterococcus faecalis*

More than 1000 different bacterial species can be found in the human oral cavity. These bacteria can form bacterial biofilm on various surfaces, and some of them are able to cause pathological changes requiring treatment. Among many others, bacterial biofilm can also be formed in the teeth root canal system, and due to canals‘ complicated anatomy with many variabilities, it is technically possible to remove present bacteria only from about 50% of the root canal wall by mechanical means only [32]. Therefore, mechanical biofilm removal needs to be accompanied by the usage of endodontic irrigants with antimicrobial properties, such as sodium hypochlorite (NaOCl), chlorhexidine gluconate, or ethylenediaminetetraacetic acid (EDTA). Despite combining mechanical methods with antimicrobial irrigants, some bacteria in the root canal system are unable to be eliminated and may result in treatment failure [33]. One of the most resistant bacteria to current endodontic disinfectants and one of the most frequently present bacteria in the endodontic biofilm is *Enterococcus faecalis*.

As mentioned above, second-generation LPPOs were investigated as an additive to the bone cement, and the DR-6155 compound demonstrated complete biofilm inhibition for all tested bacterial strains. In this study, LPPOs DR-6328 and DR-6487 LPPOs were compared to 2.5% NaOCl, 0.12% chlorhexidine digluconate, and 17% EDTA in *Enterococcus faecalis* eradication ability in simulated endodontic infection. The root canal’s dentin was used as a carrier for *Enterococcus faecalis* biofilm formation in the extracted human mature mandibular premolars. After extraction, all the premolars were disinfected, the clinical crown was removed, and after the roots were shaped, the premolars were filled with cultivation broth and 0.25% glucose. *Enterococcus faecalis* was inoculated into each tooth with the exception of negative control ones. After cultivation at 35 °C for 24 h, the broth was washed out, and the number of planktonic cells and the evaluation of the biofilm formation were assessed [34].

LPPOs were tested in two concentrations. Both LPPOs in both concentrations showed a more pronounced reduction in bacteria growth and inhibition of the biofilm formation in the root canals as compared to chlorhexidine digluconate and EDTA, but comparable with 2.5% NaOCl. NaOCl usage is, however, associated with side effects such as aggressive effects on soft and periapical tissues in the case of extrusion, allergic reactions, and unpleasant smell and taste [34]. None of these side effects was described with the use of LPPOs.

## 3. Conclusions

Since the accelerated emergence of AMR is becoming a global problem and multiresistant bacterial strains are becoming commonplace, there is a medical need for new groups of antibiotics that would have a low propensity for resistance development. LPPOs are promising antibacterial compounds that belong among small molecule membrane targeting agents. LPPOs are easy-to-be synthesized molecules that consist of four modules. By modification of these modules, different compounds with different characteristics can be created.

There are three generations of LPPOs described: First-generation LPPOs, second-generation LPPOs, and LEGO-LPPOs. All three generations have a similar mode of bactericidal action of targeting and disrupting the bacterial cytoplasmic membrane. The first generation proved to be active against Gram-positive pathogens, including the multiresistant strains. They have a limited effect on eukaryotic cells at the bactericidal concentrations, and investigators failed to select bacteria resistant to them. Disadvantages of the first generation are the inability to kill Gram-negative bacterial pathogens and the lack of activity in the presence of albumin. Therefore, it is impossible to use those compounds in systemic therapy. The second generation keeps all the advantages of the first one and, on top, broadens the antibiotic effect against Gram-negative bacteria. The second generation was investigated in local treatments as additives to the surgical bone cement, in infected wound treatment when loaded into nanofiber wound dressing, and in tooth root canal infection treatment. Results in all three indications seem to be encouraging. LEGO–LPPOs, as the most promising and evolution-wise latest generation, proved to be active against both Gram-positive and Gram-negative bacteria with equal or even better antimicrobial properties and significantly better selectivity compared to first- and second-generation resistance to LEGO–LPPOs was also not detected, and most importantly, LEGO–LPPOs keep their bactericidal activity even in the presence of serum albumin and therefore this group can be further developed as systemic therapy (see Table 1). The safety profile of LEGO–LPPOs was locally demonstrated on the skin and eye of mammals, and the maximum tolerated dose in mice was already set for further investigations.

In systemic treatment, along with the efficacy that was already proven for LEGO–LPPOs, safety and tolerability are very important aspects, and these two characteristics are often associated with the drug pharmacokinetics. Thus far, there is only limited information on how the LPPOs are metabolized and whether they interact with the cytochrome P450 enzyme system (CYP) responsible for the metabolism of xenobiotics. Hence, the interaction of a drug with the CYP may influence many other metabolic processes and the metabolism of other medicines. The drug-drug interactions through the alteration of metabolism by CYPs may lead to unexpected side effects, therapeutic failures, and, in the worst cases, may even have fatal consequences [35,36]. Therefore, the knowledge of whether the drug does or does not interact with CYP enzymes can help to minimize the possibility of adverse drug reactions as well as the interactions with other drugs concomitantly administered. Thus, the interaction of the LPPOs and the CYPs warrants further investigation. The antifungal activity of the LPPOs was not investigated, and it would also be interesting and beneficial to know it.

## Figures and Tables

**Figure 1 pharmaceutics-15-02395-f001:**
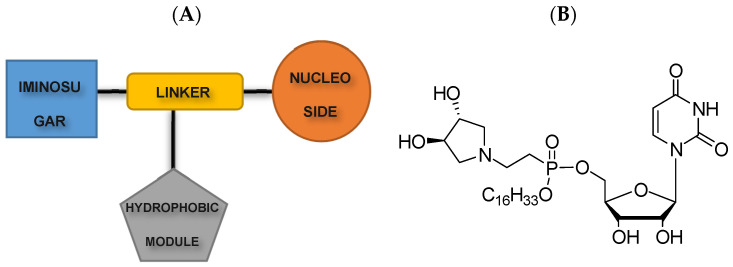
(**A**) General structure of lipophosphonoxins is composed of four modules: Iminosugar module, Linker module, Nucleoside module, and hydrophobic module; (**B**) example of the first-generation LPPO—compound DR-5026.

**Figure 2 pharmaceutics-15-02395-f002:**
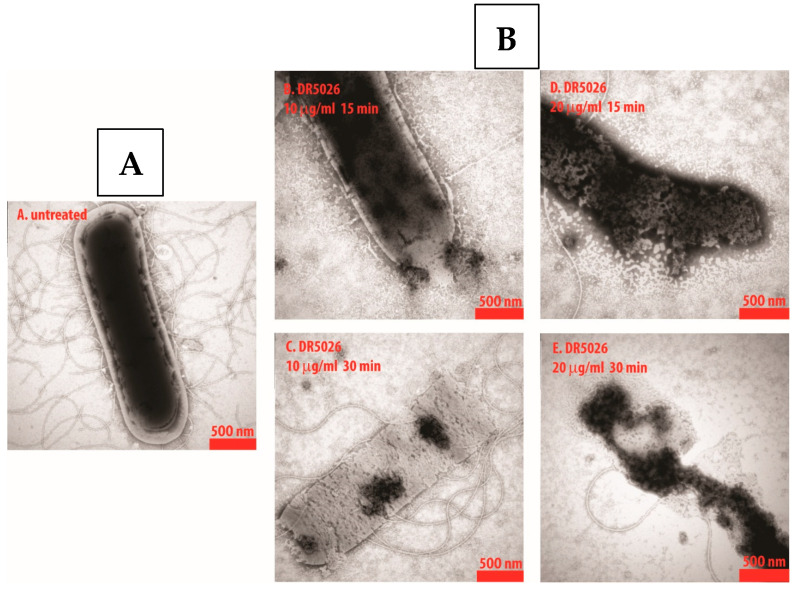
Scanning electron microscopy of *Bacillus subtilis* cells: (**A**) Control (no treatment); (**B**) Pore formation after adding first-generation LPPO DR-5026 at concentrations of 10 and 20 mg/L for 15 and 30 min.

**Figure 3 pharmaceutics-15-02395-f003:**
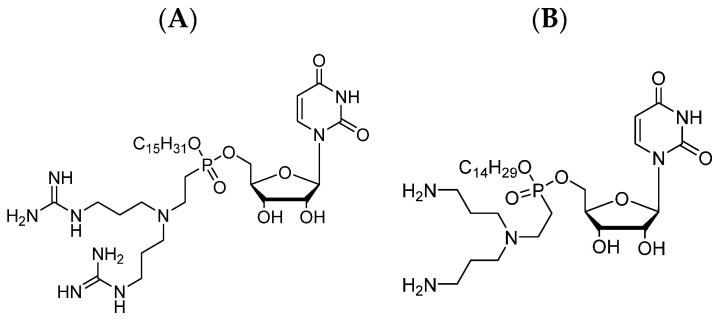
Example of the second-generation LPPOs: (**A**) Compound DR-6180; (**B**) Compound DR-6155.

**Figure 4 pharmaceutics-15-02395-f004:**
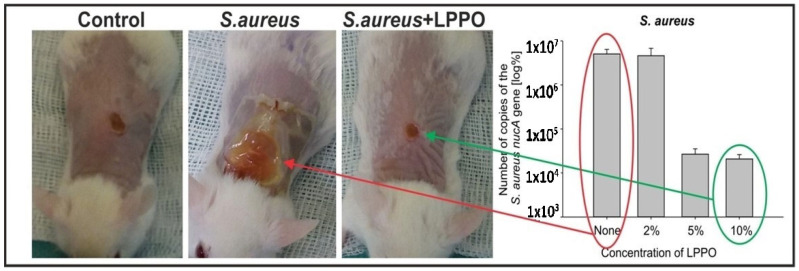
Wound healing in BALB/c mice infected with *Staphylococcus aureus* promoted by NANO–LPPO. The right panel represents qPCR quantification of *Staphylococcus aureus* (Control, NANO–LPPO2%, NANO–LPPO5%, NANO–LPPO10%).

**Figure 5 pharmaceutics-15-02395-f005:**
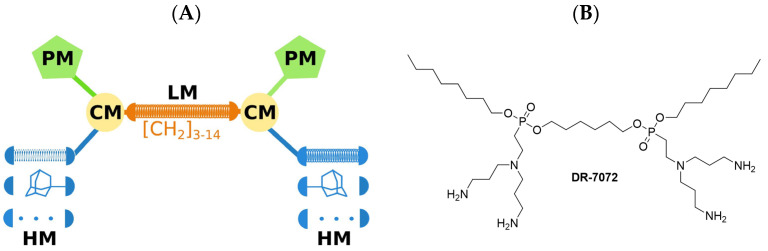
(**A**) General structure of LEGO–LPPO (PM—polar module, CM—connector module, LM—linker module, HM—hydrophobic module); (**B**) Example of LEGO–LPPO DR-7072.

**Table 1 pharmaceutics-15-02395-t001:** Comparison of three generations of LPPOs.

	Cytotoxicity against	MIC	Activity in the Presence of Albumin	Resistance Development
1st generation LPPOs	G+	1–12 mg/L	NO	not observed
2nd generation LPPOs	G+ and G−	<1 to 6 mg/L	NO	not observed
LEGO-LPPOs	G+ and G−	<1 to 8 mg/L	YES	not observed

Legend: G+ = Gram-positive pathogens; G− = Gram-negative pathogens; MIC = minimum inhibitory concentration.

## Data Availability

The data presented in this study are available in this article.

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
