# Peer review of "Lipophosphonoxins—A Novel Group of Broad Spectrum Antibacterial Compounds"

_pharmaceutics, 2023, doi:10.3390/pharmaceutics15102395_

Round 1

Reviewer 1 Report

Dear Editor, Dear Authors,

I was invited to evaluate the review « Lipophosphonoxins – a Novel Group of Broad Spectrum Antibacterial Compounds » by Viktor Šebek et al.

In this review, the authors describe and discuss findings regarding the use of Lipophosphonoxins as antimicrobial. These molecules belong to membrane-targeting antibiotics, causing lysis of the bacterial bacterial membrane with limited action on eukaryotic one. The authors also describe how the molecules were improved in term of activity and safety. The authors aim here to review and compile published data on LPPOs.

I found the review very interesting and informative.

I will have few comments to be addressed.

1- The authors may also add into their introduction informations about other membrane-targetting molecules such as polymyxin (very old use), but also Antimicrobial Peptides (AMPs) or antimicrobial peptidomimetic and polymers.

2- The authors refer to the sensitivity (or not) of the molecules to serum albumin, but what about sensitivity to NaCl ? Indeed, inhibition by NaCl has been found for various cationic amphiphillic molecules, limiting their use in vivo where NaCl is present (contrarly to MH media used for MIC where NaCl is absent or low).

3- Although they gave range of MIC in the text, it would be good for the readers to have in one table, a summary of known MIC, HC50 and IC50 (citing references) to be able to easely/quickly compare them.

4- What about the antifungal effect of the LPPOs ? Although they seems not active on eukaryotic cells, some AMPs not active on human cells are still active on yeast/fungi due to difference in lipids between humans/mammals and fungi. Please ad dit to the review or mentione it will be interesting to investigate.

5- Minor : Line 192 : « After a couple of cycles of 24 24-hour incubation and increase of concentration » please correct to « increase in » and check the full document for increase/decrease OF to be corrected to increase/decrease IN.

regards

Dear Editor, Dear Authors,

I was invited to evaluate the review « Lipophosphonoxins – a Novel Group of Broad Spectrum Antibacterial Compounds » by Viktor Šebek et al.

In this review, the authors describe and discuss findings regarding the use of Lipophosphonoxins as antimicrobial. These molecules belong to membrane-targeting antibiotics, causing lysis of the bacterial bacterial membrane with limited action on eukaryotic one. The authors also describe how the molecules were improved in term of activity and safety. The authors aim here to review and compile published data on LPPOs.

I found the review very interesting and informative.

I will have few comments to be addressed.

1- The authors may also add into their introduction informations about other membrane-targetting molecules such as polymyxin (very old use), but also Antimicrobial Peptides (AMPs) or antimicrobial peptidomimetic and polymers.

2- The authors refer to the sensitivity (or not) of the molecules to serum albumin, but what about sensitivity to NaCl ? Indeed, inhibition by NaCl has been found for various cationic amphiphillic molecules, limiting their use in vivo where NaCl is present (contrarly to MH media used for MIC where NaCl is absent or low).

3- Although they gave range of MIC in the text, it would be good for the readers to have in one table, a summary of known MIC, HC50 and IC50 (citing references) to be able to easely/quickly compare them.

4- What about the antifungal effect of the LPPOs ? Although they seems not active on eukaryotic cells, some AMPs not active on human cells are still active on yeast/fungi due to difference in lipids between humans/mammals and fungi. Please ad dit to the review or mentione it will be interesting to investigate.

5- Minor : Line 192 : « After a couple of cycles of 24 24-hour incubation and increase of concentration » please correct to « increase in » and check the full document for increase/decrease OF to be corrected to increase/decrease IN.

regards

Author Response

Dear Reviewer,

On behalf of all authors of the paper ‘Lipophosphonoxins – a Novel Group of Broad Spectrum Anti-bacterial Compounds’, I would like to sincerely thank you for your review and positive feedback.

Thank you also for your objections and please find below our comments on them.

  • The authors may also add into their introduction informations about other membrane-targetting molecules such as polymyxin (very old use), but also Antimicrobial Peptides (AMPs) or antimicrobial peptidomimetic and polymers.
  • The other 2 reviewers recommended omitting this section and focusing primarily on LPPOs, therefore the section about antimicrobial peptides was shortened and moved to Introduction.
  • The authors refer to the sensitivity (or not) of the molecules to serum albumin, but what about sensitivity to NaCl ? Indeed, inhibition by NaCl has been found for various cationic amphiphillic molecules, limiting their use in vivo where NaCl is present (contrarly to MH media used for MIC where NaCl is absent or low).
  • Thank you for this interesting note. Indeed, in none of the published papers about LPPOs, their activity in the presence of NaCl was specifically mentioned.
  • Rejman, co-author of this paper and the man behind the discovery of LPPOs, informed us that all MIC values were tested in a media with a physiological, or close to a physiological NaCl concentration.
  • Although they gave range of MIC in the text, it would be good for the readers to have in one table, a summary of known MIC, HC50 and IC50 (citing references) to be able to easely/quickly compare them.
  • Thank you for this valuable comment. A table comparing all three generations of LPPOs was added to the Conclusions.

cytotoxicity against

MIC

activity in the presence of albumin

resistance development

1st generation LPPOs

G+

1-12 mg/L

NO

not observed

2nd generation LPPOs

G+ and G-

<1 to 6 mg/L

NO

not observed

LEGO-LPPOs

G+ and G-

<1 to 8 mg/L

YES

not observed

  • What about the antifungal effect of the LPPOs ? Although they seems not active on eukaryotic cells, some AMPs not active on human cells are still active on yeast/fungi due to difference in lipids between humans/mammals and fungi. Please ad dit to the review or mentione it will be interesting to investigate.
  • Thank you for this comment. See below the sentence that was added to the Conclusions.

‘The antifungal activity of the LPPOs was not investigated and it would be also interesting and beneficial to know it.’

  • Line 603-604
  • Minor : Line 192 : « After a couple of cycles of 24 24-hour incubation and increase of concentration » please correct to « increase in » and check the full document for increase/decrease OF to be corrected to increase/decrease IN.
  • Thank you for spotting this mistake which was corrected.

In case of any other comments or questions, please do not hesitate to contact us.

Thank you and with Kind regards

Kolar M., on behalf of all authors

Reviewer 2 Report

Sebek and colleagues presented a review on lipophosphonoxins (LPPOs), a group of peptidomimetic compounds, emphasizing their antimicrobial activity and mammalian safety.

Overall, the review is well written. However, some issues can be discussed in the manuscript:

- For nonexpert on the subject, topics 2 and 2.1 seem not to be important for the review, since the classification of LPPO as a peptidomimetic is only introduced in topic 3, line 104. You can anticipate this definition in the introduction or in the abstract.

- Although it is not the aim of the review, but considering that the main target of LPPO is the cell membrane; and the authors emphasize that differences between bacterial and eukaryotic cell membranes justify the study of this group of compounds, it would be interesting to add a brief description of these differences.

- As the subject of the review are LPPOs, it is important to present the structure/name of the compounds being discussed. The description of "the most active compounds" studied is incomplete. A Table as supplementary material with the main compounds and microbial species whose antimicrobial activity was evaluated will add value to the review.

- In line 138, it is important to add which Gram-negative bacterial membrane LPPOs cannot cross.

- Second generation LPPOs also had an inhibitory effect on the outer membrane of Gram-negative bacteria?

- Line 267 – 269: “Infections are uncommon but serious complications associated with orthopedic implants and may result in the development of bacterial biofilms on the implant surface usually during the early postoperative period.” Please clarify

- Lines 356-357: “subsequently released lower dose still suffices to prevent bacterial inhibition.” Please clarify.

- Lines 392-394: “however in no sample with LPPO5% and 10% (with one exception)”. With one exception??

- Figure 4: Please improve the identification of the images. Check that on the y axis of the bar graph, the values correspond to log%. Are these images taken from published studies? Reference? “BALB/c mice” instead of “Balb/c”.

- Line 429: “Investigation of the third-generation LPPOs was a dead end”. Please clarify.

- Lines 452-454: “Antibacterial activities of LEGO-LPPOs were evaluated in Gram-positive and Gram-negative bacterial pathogens, including the resistant strains, and MIC values and hemolytic activity (HC50) were also tested”. MIC is a methodology that evaluates antibacterial activity, therefore it is redundant.

- Lines 563-565: “Disadvantages of the first-generation are the inability to kill Gram-negative bacterial pathogens and infectivity in the presence of albumin”. Infectivity? Please clarify.

- Please revise the English language.

Minor comments

-Figures 1 and 3: please add the name of the compounds.

-Line 157: please add the meaning of IC50

-Figure 2: please, add italic to “Bacillus subtilis

- Line 211: please verify if “positive change” should be “positive charge”

- Line 250: “minimal inhibitory concentration” can be changed to “MIC”

- Line 300: please add italic to “ica

“re-epithelialization” (line 372) and  “re-epithelization” (line 383) - please standardize

Reviewing the English language can improve the reading of some paragraphs, which are long and sometimes confusing.

Author Response

Dear Reviewer,

On behalf of all authors of the paper ‘Lipophosphonoxins – a Novel Group of Broad Spectrum Anti-bacterial Compounds’, I would like to sincerely thank you for your review and positive feedback.

Thank you also for your objections and please find below our comments on them.

  • For nonexpert on the subject, topics 2 and 2.1 seem not to be important for the review, since the classification of LPPO as a peptidomimetic is only introduced in topic 3, line 104. You can anticipate this definition in the introduction or in the abstract.

Based on your comment we took out sections 2 and 2.1. The part about antimicrobial peptides was shortened and added to the Introduction.

  • Although it is not the aim of the review, but considering that the main target of LPPO is the cell membrane; and the authors emphasize that differences between bacterial and eukaryotic cell membranes justify the study of this group of compounds, it would be interesting to add a brief description of these differences.

Structural differences between prokaryotic and eukaryotic membranes were added.

Whereas prokaryotic membranes have negatively charged lipids and cholesterol is universally absent there, the eukaryotic membranes are composed of phospholipids with neutral charge and cholesterol, which increases the membrane thickness and decreases its fluidity, is present there [21]. This structural difference between the membranes explains why LPPOs don’t cross the eukaryotic plasmatic membrane at their bactericidal concentrations and can’t create pores in it.

  • Lines 150-155

Reference: Glukhov, E.; Stark, M.; Burrows, L. L. et al. Basis for selectivity of cationic antimicrobial peptides for bacterial versus mammalian membranes. J. Biol. Chem. 2005, 280, 33960−33967

  • As the subject of the review are LPPOs, it is important to present the structure/name of the compounds being discussed. The description of "the most active compounds" studied is incomplete. A Table as supplementary material with the main compounds and microbial species whose antimicrobial activity was evaluated will add value to the review.

The structures of LPPOs discussed in the article are shown in the Figures (Fig. 1, 3, and 5) in respective chapters now.

  • In line 138, it is important to add which Gram-negative bacterial membrane LPPOs cannot cross.

The text was corrected:

The lack of efficacy against Gram-negative bacteria is likely due to LPPOs’ inability to cross both bacterial membranes, outer membrane and cytoplasmatic membrane.

  • Lines: 132-134

  • Second generation LPPOs also had an inhibitory effect on the outer membrane of Gram-negative bacteria?

Yes. The second-generation LPPOs have higher positive change due to their modified polar module which enables them to disrupt also the outer membrane of the Gram-negative bacteria and form the pores.

  • Line 267 – 269: “Infections are uncommon but serious complications associated with orthopedic implants and may result in the development of bacterial biofilms on the implant surface usually during the early postoperative period.” Please clarify

The text was corrected:

Infections are serious complications associated with 1%-2% of primary joint replacements and may result in the development of bacterial biofilms on the implant surface that can result in implant removal [23].

  • Lines: 271-273

Reference: 23. Fernandes, A.; Dias, M. The microbiological profiles of infected prosthetic implants with an emphasis on the organisms which form biofilms. J Clin Diagn Res. 2013;7:219–223.

  • Lines 356-357: “subsequently released lower dose still suffices to prevent bacterial inhibition.” Please clarify.

The text was corrected:

LPPO is released in two phases. First comes an initial spike that lasts for a few hours and is followed by a gradual lower release in subsequent days. This biphasic release is optimal for local infection treatment when elevated initial concentration kills present bacteria and subsequently released lower dose still suffices to prevent new bacterial infection from occurring.

  • Lines 298-302

  • Lines 392-394: “however in no sample with LPPO5% and 10% (with one exception)”. With one exception??

The text was corrected:

On agar media, Staphylococcus aureus was detected in all samples without LPPO and with NANO-LPPO2% dressing, however only in one sample with LPPO5%.

  • lines 397-399
  • Figure 4: Please improve the identification of the images. Check that on the y axis of the bar graph, the values correspond to log%. Are these images taken from published studies? Reference? “BALB/c mice” instead of “Balb/c”.

The photos were taken during the experiment and the photos were shared with us by the investigation team. Photos were not published before.

The initial experiment is referenced – reference number 29.

“BALB/c” in Figure 4 was corrected. 

  • Line 429: “Investigation of the third-generation LPPOs was a dead end”. Please clarify.

This paragraph was added:

To overcome the limitation of second-generation LPPOs, structure−activity relationship (SAR) studies were performed aiming to synthesize the next-generation LPPO that would keep excellent antimicrobial effects even in the presence of serum albumin and thus make it suitable for systemic treatment. Modifying various modules of the second-generation LPPOs (e.g. the nucleoside module) did not lead to significant improvement. Although some of the synthesized compounds retained their antibacterial activity, this ability was also lost in the presence of bovine serum albumin [14]. Thus, a new structural scaffold termed LEGO-LPPO was introduced and if third-generation LPPO collocation is used, it refers to LEGO-LPPOs today.

  • Line 431-439

  • Lines 452-454: “Antibacterial activities of LEGO-LPPOs were evaluated in Gram-positive and Gram-negative bacterial pathogens, including the resistant strains, and MIC values and hemolytic activity (HC50) were also tested”. MIC is a methodology that evaluates antibacterial activity, therefore it is redundant.

Corrected

Antibacterial activities of LEGO-LPPOs were evaluated in Gram-positive and Gram-negative bacterial pathogens, including the resistant strains, and hemolytic activity (HC50) was also tested.

  • Line 455-457

  • Lines 563-565: “Disadvantages of the first-generation are the inability to kill Gram-negative bacterial pathogens and infectivity in the presence of albumin”. Infectivity? Please clarify.

Corrected

Disadvantages of the first-generation are the inability to kill Gram-negative bacterial pathogens and the lack of activity in the presence of albumin…..

  • Lines 568-570

- Please revise the English language.

English was revised and typos were corrected.

Minor comments

-Figures 1 and 3: please add the name of the compounds.

            - added

-Line 157: please add the meaning of IC50

            - added, line 159

-Figure 2: please, add italic to “Bacillus subtilis”

            - corrected

- Line 211: please verify if “positive change” should be “positive charge”

            - corrected, line 213

- Line 250: “minimal inhibitory concentration” can be changed to “MIC”

            - corrected

- Line 300: please add italic to “ica”

            - corrected

-“re-epithelialization” (line 372) and  “re-epithelization” (line 383) - please standardize

  • Corrected, “re-epithelization” in both cases, line 378 and 389

In case of any other comments or questions, please do not hesitate to contact us.

Thank you and with Kind regards

Kolar M., on behalf of all authors

Reviewer 3 Report

Lipophosphonoxins (LPPOs) represent a new group of membrane-targeting antibiotics.The authors summarized and compared the antibacterial activities of three generations of LPPOs, and looked forward to the future application and development of LPPOs, which would help to understand this kind of antibacterial agent and discover more corresponding antibacterial drug. But I have several following concerns:

1. It is recommended to draw a schematic diagram of the antimicrobial effects of LPPOs.

2. The focus of the full text is LPPOs, why did the author describe AMPs in a separated section? And this section has only one subsection, it is no neccesdary to use secondary headings.

3. Representative compounds of the three generations of LPPOs, their structures, and antimicrobial profiles should be listed to assist the reader in quickly, effectively, and intuitively understanding the antimicrobial activities of this class of compounds.

4. Abbreviations should be defined when they first appear in the text.

5. Latin species names should be italicized, such as "Bacillus subtilis" on line 162.

6. More detailed legend should be added to each Figures so that the reader can understand roughly what the author is saying without looking at the text.

7. It is suggested to use a more intuitive table to compare the advantages and disadvantages of the three generations of LPPOs.

8.  Please unify the format of references in the article, including the author's name, the case of words in the title of the article, the writing of the name of the journal, and the page number.

Minor editing of English language required.

Author Response

Dear Reviewer,

On behalf of all authors of the paper ‘Lipophosphonoxins – a Novel Group of Broad Spectrum Anti-bacterial Compounds’, I would like to sincerely thank you for your review and positive feedback.

Thank you also for your objections and please find below our comments on them.

  • It is recommended to draw a schematic diagram of the antimicrobial effects of LPPOs.

  • Thank you for your suggestion, but the authors would rather prefer the show the mode of action of LPPOs on the scanning electron microscopy photos, see Figure 2. We think that this way of MoA depiction might be as demonstrative as the drawing.

  • The focus of the full text is LPPOs, why did the author describe AMPs in a separated section? And this section has only one subsection, it is no neccesdary to use secondary headings.

  • Based on your comment we took out sections 2 and 2.1. The part about antimicrobial peptides was shortened and added to the Introduction.

  • Representative compounds of the three generations of LPPOs, their structures, and antimicrobial profiles should be listed to assist the reader in quickly, effectively, and intuitively understanding the antimicrobial activities of this class of compounds.

  • Thank you for this valuable comment. A table comparing all three generations of LPPOs was added to the Conclusions.

cytotoxicity against

MIC

activity in the presence of albumin

resistance development

1st generation LPPOs

G+

1-12 mg/L

NO

not observed

2nd generation LPPOs

G+ and G-

<1 to 6 mg/L

NO

not observed

LEGO-LPPOs

G+ and G-

<1 to 8 mg/L

YES

not observed

  • Abbreviations should be defined when they first appear in the text. Corrected

  • Latin species names should be italicized, such as "Bacillus subtilis" on line 162.
  • Corrected

  • More detailed legend should be added to each Figures so that the reader can understand roughly what the author is saying without looking at the text.
  • More detailed legends added to all Figures

  • It is suggested to use a more intuitive table to compare the advantages and disadvantages of the three generations of LPPOs.
  • See the table at objection number 3)

  • Please unify the format of references in the article, including the author's name, the case of words in the title of the article, the writing of the name of the journal, and the page number.
  • All references were reviewed and edited.

In case of any other comments or questions, please do not hesitate to contact us.

Thank you and with Kind regards

Kolar M., on behalf of all authors

Round 2

Reviewer 3 Report

The authors have addressed all my concerns. I recommend accept it in current status.